# The Function of Autophagy as a Regulator of Melanin Homeostasis

**DOI:** 10.3390/cells11132085

**Published:** 2022-06-30

**Authors:** Ki Won Lee, Minju Kim, Si Hyeon Lee, Kwang Dong Kim

**Affiliations:** 1PMBBRC, Gyeongsang National University, Jinju 52828, Korea; leemaskup@naver.com; 2Division of Applied Life Science, Gyeongsang National University, Jinju 52828, Korea; kmj941226@naver.com (M.K.); 2tlgus@naver.com (S.H.L.)

**Keywords:** autophagy, melanogenesis, melanin

## Abstract

Melanosomes are melanocyte-specific organelles that protect cells from ultraviolet (UV)-induced deoxyribonucleic acid damage through the production and accumulation of melanin and are transferred from melanocytes to keratinocytes. The relatively well-known process by which melanin is synthesized from melanocytes is known as melanogenesis. The relationship between melanogenesis and autophagy is attracting the attention of researchers because proteins associated with autophagy, such as WD repeat domain phosphoinositide-interacting protein 1, microtubule-associated protein 1 light chain 3, autophagy-related (ATG)7, ATG4, beclin-1, and UV-radiation resistance-associated gene, contribute to the melanogenesis signaling pathway. Additionally, there are reports that some compounds used as whitening cosmetics materials induce skin depigmentation through autophagy. Thus, the possibility that autophagy is involved in the removal of melanin has been suggested. To date, however, there is a lack of data on melanosome autophagy and its underlying mechanism. This review highlights the importance of autophagy in melanin homeostasis by providing an overview of melanogenesis, autophagy, the autophagy machinery involved in melanogenesis, and natural compounds that induce autophagy-mediated depigmentation.

## 1. Introduction

The process by which melanin, the pigment of the skin, is synthesized in melanocytes is known as melanogenesis [1,2]. Melanogenesis is induced via various internal or external factors, such as aging, hormonal changes, and ultraviolet (UV) B-mediated skin irritation, and uncontrolled melanogenesis causes hyperpigmentation of the skin, which leads to skin effects such as melasma, freckles, age spots, and dark spots [3,4]. The melanosome is a melanocyte-specific lysosome-related organelle in which melanin pigment is synthesized and stored [5], and melanosomes are transferred from melanocytes to keratinocytes [6]. Conversely, the hypopigmentation that occurs in vitiligo is usually a result of inflammation caused by skin stress or due to other causes, including innervation, microvascular malformation, degeneration of melanocytes by oxidative stress, adhesion defects of melanocytes, somatic mosaic, and genetic influences [5,7,8,9,10,11,12,13].

Particularly, researchers have proposed that autophagy may play a role in redox stress-related vitiligo [14,15]. Impairment of autophagy may disrupt the antioxidant defense system, causing oxidative damage to melanocytes. Autophagy is a highly conserved cellular degradation and recycling process in all eukaryotes, and three types of autophagy occur in mammalian cells: microautophagy, macroautophagy, and chaperone-mediated autophagy. Although each type is morphologically distinct, all three have in common the delivery of cargo to lysosomes for degradation and recycling [16]. Among these three autophagy types, macroautophagy has been well-studied and is known to play an important role in maintaining intracellular homeostasis by inducing the degradation of cytoplasmic substances or metabolites under stress conditions, e.g., nutrient or energy deprivation, and decomposition of damaged or unnecessary organelles [16,17]. Interestingly, some research has shown that autophagy is an important factor in determining skin color. Caucasian skin-derived keratinocytes exhibit higher autophagic activity than those derived from African-American skin, and the accumulation of melanosomes is known to be accelerated via treatment with lysosomal inhibitors or small interfering ribonucleic acids specific to autophagy-associated proteins [18]. Additionally, various studies have provided evidence that the autophagy machinery may regulate melanogenesis.

In this review, we discuss the signal transduction pathways that induce melanogenesis, the relationship between the autophagy machinery and melanogenesis, and autophagy-inducing skin whitening materials.

## 2. Signal Transduction Pathways That Induce Melanogenesis

The synthesis of melanin in melanosomes is the result of complex pathways involving enzyme reactions. Tyrosinase (TYR), tyrosine-related protein-1 (TRP-1), and TRP-2 are mainly involved in enzyme reactions that transform tyrosine to melanin pigments [19]. Figure 1 illustrates common signaling pathways inducing melanogenesis.

Melanogenesis can be induced by various factors, including adrenocorticotropic hormone (ACTH) [20], α-melanocyte-stimulating hormone (α-MSH) [21,22], and stem cell factor (SCF) [23,24]. These factors induce melanogenesis through microphthalmia-associated transcription factor (MITF) expression and activation, which in turn induces the expression of pigment-related genes, such as *TYR*, *TRP-1*, *TRP-2*, and premelanosome protein (*PMEL*) [25]. Melanocortin 1 receptor (MC1R) is expressed in melanocytes in the plasma membrane, and ACTH and α-MSH are the ligands of MC1R [26]. MC1R-mediated signaling induces adenosine 3′,5′-cyclic monophosphate (cAMP), which activates PKA [27,28,29]. Activated PKA translocates into the nucleus and phosphorylates cAMP-response element-binding protein (CREB) [30]. CREB co-operates with SOX10 to induce MITF expression, resulting in the expression of pigment-related genes [31,32]. The binding of SCF to its receptor, tyrosine-protein kinase kit (c-KIT), initiates mitogen-activated protein kinase (MAPK) cascades that induce melanogenesis [33]. Autophosphorylated c-KIT activates p38 MAPK, resulting in CREB phosphorylation and sequential MITF activation [34,35,36]. The SCF/c-KIT pathway also activates extracellular signal-regulated kinase (ERK), inducing CREB phosphorylation for melanogenesis, whereas Ser73 phosphorylation of MITF via ERK leads to proteasomal degradation of MITF [37,38]. Furthermore, the SCF/c-KIT pathway is associated with phosphoinositide 3-kinase (PI3K) signaling that leads to glycogen synthase kinase-3 β (GSK3β) inactivation, which contributes to increasing the stability of β-catenin operating as a cofactor for MITF [39,40]. Wnt signaling is also a representative signaling pathway that contributes to β-catenin stability and plays a role in melanogenesis [41,42,43,44,45]. Frizzled-1 as a receptor for Wnt couples via G proteins, Go and Gq, and Dvl to activate β-catenin [46].

## 3. Autophagy

Starvation is a representative type of stress that induces macroautophagy, which occurs through sequential events involving initiation, nucleation, elongation and substrate selection, fusion of the autophagosome and lysosome, and lysosomal degradation [47,48]. During cell starvation, the lack of nutrients increases cellular 5′-adenosine monophosphate (AMP) levels. The ratio of AMP to ATP leads to AMP-activated protein kinase (AMPK) activation and inactivation of the target of the rapamycin complex 1 (mTORC1) [16,49], resulting in the activation of an autophagy-initiation complex containing FIP200, ULK1, autophagy-related (ATG)101, and ATG13 [50]. Under rapamycin treatment or starvation, mTORC1 is dissociated from the initiation complex, ATG13 and ULK1/2 become partially dephosphorylated, and autophagy is induced [51,52]. Phagophore nucleation is triggered upon phosphatidylinositol 3-phosphate (PI3P) generation by a complex with class III PI3K activity consisting of VPS34, VPS15, beclin-1, AMBRA1, and/or UV-radiation resistance-associated gene (UVRAG) along with the recruitment of vesicles containing ATG9 [53,54,55,56,57]. Elongation of phagophores formed with the support of WD repeat domain phosphoinositide-interacting protein (WIPI) includes two ubiquitin-like conjugation systems. ATG7 and ATG10 operate sequentially to catalyze the formation of the ATG12–ATG5:ATG16L1 complex. ATG4, ATG7, and ATG3 function together to cut the precursors of microtubule-associated protein 1 light chain 3 (LC3)-like proteins into their mature forms, after which they bond to phosphatidylethanolamine to generate LC3II-B, which is recruited and integrated into growing phagophores in an Atg5–Atg12:ATG16L1-dependent manner [58,59,60,61,62]. Cargo and/or cargo-selective proteins allow the formation of autophagosomes by binding to LC3 and LC3 homologs. Selective autophagy that degrades specific cargoes has also been reported, although autophagy induced by starvation is nonselective. Various cargo-selective proteins, also known as autophagy receptors, recognize the ubiquitinated cargoes and mediate autophagosome formation by surrounding the cargoes through LC3II-B binding on phagophores [63]. After complete fusion of the extended ends of the phagophore membrane, the formed autophagosomes fuse with lysosomes to form autolysosomes, in which substrate degradation is mediated through luminal acidification and lysosomal hydrolases. Figure 2 illustrates the representative common autophagy process.

## 4. Autophagy Machinery That Regulates Melanogenesis

Melanogenesis in pigment cells proceeds in three stages: (1) melanogenic gene expression, (2) melanosome biogenesis and maturation, and (3) melanosome migration to the cell tip. Studies have indicated that autophagy machinery proteins may be involved in melanogenesis regulation (Figure 3) [64].

MITF, a master regulator of melanogenesis, plays an important role in melanogenesis stage 1 (i.e., the melanogenesis gene expression stage). MITF induces the expression of various genes involved in melanogenesis, such as *TYR*, *TRP-1*, *TRP-2*, and *PMEL* [65,66,67]. *ATG7*, a critical gene associated with LC3 lipidation, might also be involved in melanogenesis [68]. Knockdown of *ATG7* in natural human epidermal melanocytes decreased the MITF expression level and reduced melanin accumulation in the cells, whereas overexpression of ATG7 increased MITF expression [69]. Additionally, tail skin pigmentation in ATG7^f/f^Tyr:Cre mice was consistently lower than that in ATG7^f/f^ mice [70]. In Melan-a cells, knockdown of LC3 decreased ERK activity, which suppressed α-MSH-mediated melanogenesis by attenuating phosphorylation of CREB and MITF expression [71]. Moreover, mice heterozygous for beclin-1, a scaffold protein in the class III PI3K complex, showed mislocalization of MITF-nucleus, resulting in depigmentation in MNT-1 cells in relation to coat color [64], and the embryos of beclin-1-depleted zebrafish showed almost 50% lower melanin levels compared with those in control embryos. Transactional downregulation of both TYR and TRP-1 was also shown in beclin-1-depleted zebrafish [72]. WIPI1 has been reported to bind with phosphoatidylinositol-3 phosphate in the early stage autophagosome to recruit the ATG12–ATG5:ATG16L complex and elongate the autophagosome membrane [73]. In MNT-1 cells, WIPI1 induces AKT activation through activation of mTORC2, which results in the inactivation of GSK3β. WIPI1-mediated GSK3β inactivation increases β-catenin stability, which in turn induces MITF expression [74]. However, it has been suggested that ULK1, which contributes to autophagy through the activation of the class III PI3K complex, plays a role in inhibiting melanin synthesis. In one study, ULK1 knockdown in MNT-1 cells increased MITF expression, resulting in upregulation of melanogenesis (Figure 3a) [75].

Numerous proteins are involved in melanogenesis stage II, i.e., the melanosome biogenesis and maturation stage, including adaptor protein (AP)-1, AP-2, biogenesis of lysosome-related organelle complex (BLOC)-1, BLOC-2, BLOC-3, and various Rab GTPases [76,77,78,79,80]. UVRAG, identified as a beclin-1-binding autophagy-associated protein [81,82], has specialized functions in melanosome biosynthesis through its interaction with BLOC-1. UVRAG facilitates the classification and delivery of melanogenic cargoes by maintaining the localization and stability of BLOC-1. When UVRAG levels are reduced, cells do not respond to UVR–α-MSH–MITF signaling and melanocyte development becomes defective in vivo (Figure 3b) [83].

At melanogenesis stage III (i.e., melanosome movement toward the cell tip), the melanosome should move from the perinucleus to the tip of melanocytes and transfer to keratinocytes, after which the transferred melanosome plays a protective role against UV-mediated DNA damage [6]. It has been proposed that autophagic proteins, such as LC3B and ATG4, mediate melanosome trafficking in the cytoskeletal track [84]. In LC3B-knockdown B16 cells, melanosomes do not interact with microtubules and remain at the perinuclear site. LC3B lipidation and delipidation were mediated by ATG4B, and the LC3BII delipidation activity of ATG4B was critical for melanosome separation from microtubules to actin filaments (Figure 3c).

## 5. Autophagy Inducers That Induce Skin Depigmentation

While autophagy-associated proteins have been reported to play critical roles in melanogenesis in many studies, some reports suggest that autophagy-inducing agents are involved in skin depigmentation. Autophagy activity in Caucasian melanocytes was higher than that in African-American melanocytes, and decreased autophagy was shown in hyperpigmented skin, such as that in senile lentigo [18,85]. A lysosomal protease, cathepsin L, was found to be involved in melanosome degradation in melanocytes through autophagosome–lysosome fusion [86]. Some studies suggest that autophagy inducers might induce melanosome degradation in an autophagy-dependent manner (Table 1). The listed autophagy inducer-mediated depigmentation is inhibited by knockdown of autophagy essential genes, such as *ATG5* [87,88,89,90], *ATG7* [91], and *LC3* [92], or treatment with autophagy inhibitors, such as 3-MA [88,92,93,94,95], hydroxychloroquine [96], bafilomycin A1 [89], and chloroquine [97]. For example, β-mangostin cannot induce autophagy in B16F10 cells, but it induces depigmentation through autophagy-mediated melanosome degradation in pigmented B16F10 cells via α-MSH stimulation, and the depigmentation is inhibited by ATG5 knockdown or 3-MA treatment [88]. Although many autophagy machinery components are associated with melanogenesis, the degradation of existing melanosomes might be induced by autophagy. To date, the studies on agents that induce autophagy-associated skin discoloration are limited to the inhibition of depigmentation by either autophagy-associated gene knockdown or the use of autophagy inhibitors, and the molecular mechanisms underlying melanosome-targeted autophagy are yet to be clarified.

## 6. Conclusions and Outlook

In skin pigmentation homeostasis, the balance between melanogenesis and melanosome degradation is likely important. Although the melanogenesis pathway has been well-studied, and proteins that play roles in autophagy are known to be involved in melanogenesis, autophagy by starvation does not induce melanogenesis [74], and ATG7-dependent autophagy activity has little effect on melanogenesis [70]. Melanogenesis is a process leading to the synthesis of melanin in melanosome including melanosome formation, and autophagy is a process degrading cellular components. Strangely, proteins essential for autophagy degrading cellular components including melanosome are involved in melanogenesis for de novo synthesis of melanin and melanosome. Although the mechanism to clarify the relationship between the two processes has not been elucidated so far, there is no direct evidence that the autophagy process is essential for melanogenesis. Another possibility is that proteins involved in autophagy may play a role in the signaling for melanogenesis independent of the autophagy process. For example, although knockdown or knockout of *ATG7*, an essential gene for autophagy, inhibits melanin synthesis through reduction of MITF expression [69,70], knockdown of ULK1, an essential kinase forming autophagy initiation complex, induces melanin synthesis by increasing the expression of MITF [75]. Therefore, the precise role of autophagy in melanogenesis regulation must be determined as well as how the autophagy pathway cross-talks with the melanosome pathway and how the roles of several factors are balanced in various physiological processes.

It has been proposed that the autophagy process is involved in melanin degradation, particularly in skin whitening material research. Ho et al. suggested that, under stress conditions, including starvation or defective melanosomes, autophagy could be activated to form autophagosomes that engulf and degrade melanosomes [101]. Selective autophagy is an important cellular event that maintains cellular physiological homeostasis through the degradation of specific cellular compartments, such as aggregated proteins, damaged organelles, and pathogens. Selective autophagy is mediated using autophagy receptors that recognize target cargo via binding to ubiquitinated organelles. For example, membrane proteins of damaged mitochondria are recognized by PINK/PARKIN and sequentially phosphorylated and ubiquitinated. The autophagy receptor optineurin recognizes and binds to membrane proteins and links to the LC3-embedded phagophore [102]. Melanosomes are also known as specific organelles of melanocytes. The results of several studies suggest that autophagy-mediated degradation of melanosomes exists, but direct evidence and the underlying molecular mechanism have not been reported. We suggest that future research should be focused on the following: (1) how melanosomes induce the autophagy-initiation signal, (2) identification of the E3-ligase that ubiquitinates melanosome membrane proteins, and (3) identification of melanosome-targeted autophagy receptors and molecular mechanisms.

## Figures and Tables

**Figure 1 cells-11-02085-f001:**
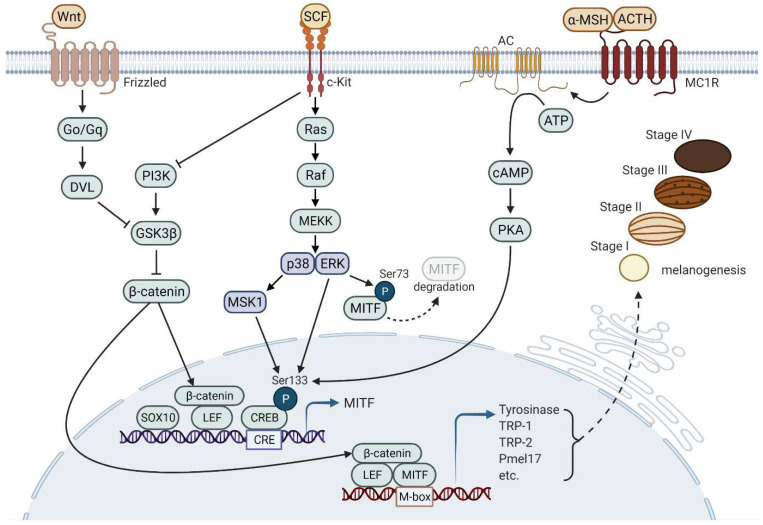
Signaling pathways that induce melanogenesis. Three representative signaling pathways, including MC1R-mediated signaling, SCF/c-KIT signaling, and Wnt signaling, are involved in melanogenesis. Expression and activation of MITF induce the expression of various proteins that play important roles in the formation and maturation of melanosomes as well as melanin synthesis. The continuous reaction of enzymatic proteins (e.g., TYR and TRP-1/2) and structural proteins (e.g., Pmel17) leads to melanogenesis in melanosomes wherein melanin pigments are synthesized and stored. DVL, Disheveled; Go/Gq, main families of G proteins.

**Figure 2 cells-11-02085-f002:**
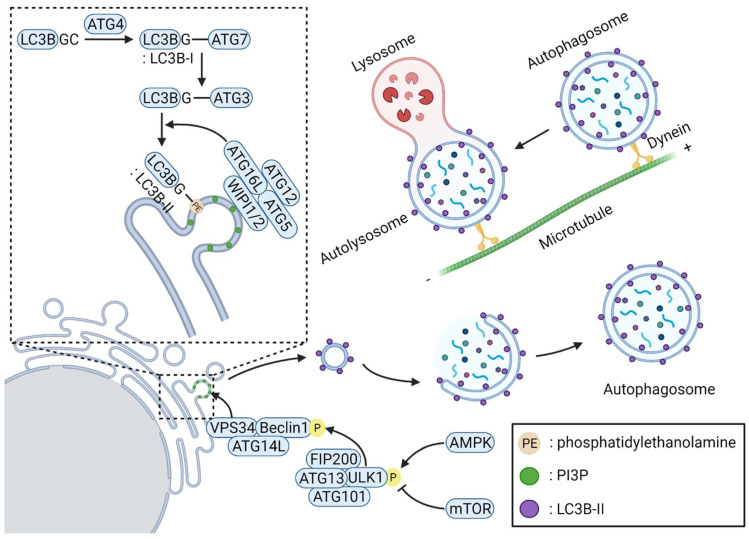
The most common autophagy process. Under nutrient-deficient conditions, increased AMP levels induce AMPK activation and AMPK-mediated activation of ULK1 complex (FIP200–ULK1–ATG13-ATG101). The ULK1 complex phosphorylates beclin-1, enabling the formation of the class III PI3K complex (VPS34–beclin-1–ATG14L). WIPI1/2 are recruited at the phagophore nucleation site by binding with PI3P, which is generated by the class III PI3K complex. ATG4 protease cleaves the C-terminal end of LC3B to expose glycine (LC3B-I). LC3B-I is then incorporated into the phagophore nucleation membrane through lipidation with phosphatidylethanolamine (LC3B-II) via sequential interactions with ATG7, ATG3, and the ATG12–ATG5:ATG16L complex. The LC3B-II-positive phagophore is then elongated and forms an autophagosome. While elongated, the random cytosolic contents are captured in the autophagosome and degraded after the autophagosome matures to an autolysosome.

**Figure 3 cells-11-02085-f003:**
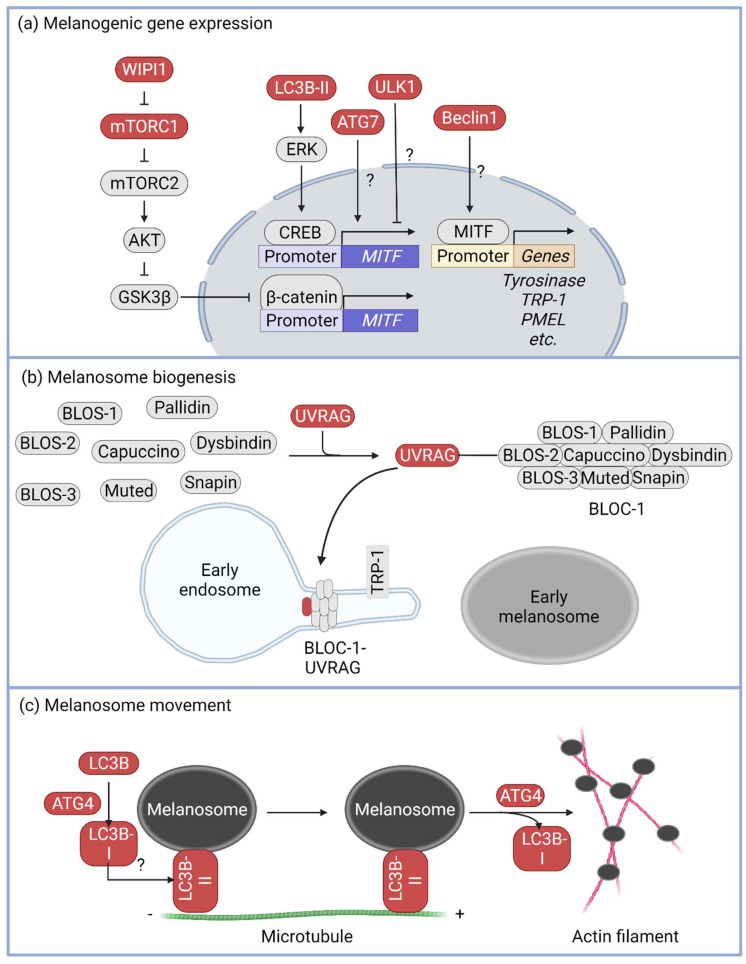
Autophagy machinery in melanogenesis. Autophagy machinery is related to melanogenesis in various steps. (**a**) In step 1, the melanogenesis-related gene expression step, *MITF* expression is regulated by WIPI1, LC3B-II, and ATG7. WIPI1 increases *MITF* expression through upregulation of β-catenin stability via GSK3β inhibition. LC3B-II induces ERK activation, and ERK increases *MITF* expression via phosphorylating CREB, whereas ATG7 and beclin-1 are positively related to *MITF* expression and MITF transcription activity, respectively, and ULK1 plays a negative role in melanogenesis. However, the precise mechanisms underlying these processes remain unclear. (**b**) In step 2, the melanosome biogenesis, UVRAG interacts with the BLOC-1 complex and upregulates its protein stability. (**c**) In step 3, the melanosome movement, LC3B is incorporated into the melanosomal membrane via cleavage through ATG4. LC3B on the melanosomal membrane mediates melanosome–microtubule interactions to facilitate melanosome movement to the cell tip. Before the transfer of the melanosome to an actin filament, lipidated LC3B on the melanosomal membrane is removed via ATG4 protease.

**Table 1 cells-11-02085-t001:** Agents that induce skin depigmentation in an autophagy-dependent manner.

Agent/Stimulation	Reported Finding	Ref.
ARP101	ATG5 knockdown inhibited the antimelanogenic effect of ARP101. Electron microscopy analysis showed that autophagosomes engulf melanosomes.	[87]
Ellagic acid (EA)	3-MA treatment or LC3 silencing significantly reduced EA-induced antimelanogenic activity in B16F10 cells.	[92]
3-O-Glyceryl-2-O-hexyl ascorbate (VC-HG)	VC-HG activates autophagy, and VC-HG-mediated depigmentation is partially inhibited by autophagy inhibitors, namely hydroxychloroquine or pepstatin A, in B16 cells.	[96]
3′-Hydroxydaidzein (3′-ODI)	3′-ODI significantly reduced α-MSH-mediated melanogenesis, and the inhibition of autophagy significantly reduced the antimelanogenic effects of 3′-ODI in α-MSH-stimulated melanoma cells.	[93]
Isoliquiritigenin	Autophagy inhibition via si-ATG7 or 3-MA treatment decreased LC3 II protein levels and increased PMEL17, p62, and melanin levels in HaCaT cells.	[91]
β-mangostin	Melanosome-engulfing autophagosomes were observed via transmission electron microscopy. Previously formed melanin could be degraded effectively in an autophagy-dependent manner, which was inhibited by ATG5 knockdown or 3-MA treatment in β-mangostin-treated B16F10 cells.	[88]
Melasolv	Melasolv suppressed the accumulation of melanin content and induced autophagy. ATG5 knockdown or bafilomycin A1 treatment suppressed melasolv-mediated depigmentation in B16F1 cells.	[89]
5-Methyl-3-tetradecylidene-dihydro-furan-2-one (DMF02)	DMF02 induced melanosome degradation via autophagy in vitro, and this degradation was inhibited by a lysosomal inhibitor, chloroquine, in B16F10 cells.	[97]
*Patrinia villosa* (Thunb.) Juss extract (Pv-EE)	Pv-EE induced autophagy, and the Pv-EE-mediated antimelanogenic effect was inhibited by 3-MA in B16F10 cells.	[94]
PTPD-12	PTPD-12 induced melanosome degradation through stimulation of autophagic flux in human melanocytes and keratinocytes.	[98]
Radiofrequency (RF) irradiation	RF irradiation upregulated autophagy-initiation factors, such as FIP200, ULK1, ULK2, ATG13, and ATG101, in the skin. Beclin-1 expression and the expression ratio of LC3-I to LC3-II increased with UV-B/RF irradiation, and melanin-containing autophagosome levels increased with RF irradiation.	[99]
Resveratrol (RSV)	ATG5 knockdown significantly suppressed RSV-mediated antimelanogenesis as well as RSV-induced autophagy in Melan-A cells.	[90]
Schaftoside	Schaftoside treatment had an antimelanogenic effect and induced autophagy activation in B16F1 cells, and 3-MA treatment reduced the antimelanogenic effect via schaftoside in B16F1 cells.	[95]
Tranexamic acid (TXA)	TXA reduced melanin accumulation by activating ERK signaling and the autophagy system.	[100]

## Data Availability

Not applicable.

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
