# Peer review of "The Function of Autophagy as a Regulator of Melanin Homeostasis"

_cells, 2022, doi:10.3390/cells11132085_

Round 1
Reviewer 1 Report
This review demonstrates a possible role of autophagy as a regulator of melanin homeostasis. Several signaling pathways are described in terms of melanogenesis. In figure 1, Wnt/Frizzled pathway includes Go/Gq protein in the cascade, which is quite interesting and some references may be added.
Table 1 needs the title to indicate that it is a list of autophagy inducers or inhibitors.
Figures may be cited in the text.
Author Response
Comment 1. In figure 1, Wnt/Frizzled pathway includes Go/Gq protein in the cascade, which is quite interesting and some references may be added.
Answer) Thank you for your advice. A reference and a sentence “Frizzled-1 as a receptor for Wnt couples via G proteins, Go and Gq, and Dvl to activate b-catenin [46].” are added, line 85, page 2.
Comment 2. Table 1 needs the title to indicate that it is a list of autophagy inducers or inhibitors.
Answer) Thank you for your kindness. Perhaps there was a mistake during the editing. We corrected that mistake, line 218, page 7.
Comment 3. Figures may be cited in the text.
Answer) Thank you for your kindness. We cited figures in the text. Figure 1 in line 62, page 2. Figure 2 in line 123, page 4. Fig3 in line 141, page 5. Fig3.1 in line 165, page 5. Fig 3.2 in line 173, page 5. Fig 3.3 in line 183, page 5.
Reviewer 2 Report
The manuscript by Ki Won Lee and co-authors reviewed the published papers on autophagy in melanin homeostasis.
The work is interesting and the authors provided an overview of mechanisms involved in the autophagy in melanogenesis, also reporting the natural compounds that induce autophagy-mediated depigmentation.
However, the manuscript must be improved. All sections of the paper must e significantly revised. In many cases, the authors have listed the overall molecules involved in the autophagic process without discussion or correlation between studies and especially without correlation with the pathology.
I suggest the authors revise the manuscript to
-i) include more schemes of molecular mechanisms described in the text;
ii) improve the section about skin depigmentation;
iii) improve the section related to natural compounds that induce autophagy-mediated depigmentation;
iv) improve the discussion among the different results presented.
Author Response
Comment 1. include more schemes of molecular mechanisms described in the text;
Answer) Thank you for your points. “phosphatidylinositol 3-phosphate (PI3P)” was added, line 105, page 3. “ATG101” was added, line 128, page 4. “Whereas, ATG7 and beclin-1 are positively related to MITF expression and MITF transcription activity respectively, ULK1 plays a negative role in melanogenesis” was added, line 188-189, page 6. ULK1 was added in Figure 3.1 melanogenic gene expression.
Comment 2 & 3. improve the section about skin depigmentation; improve the section related to natural compounds that induce autophagy-mediated depigmentation;
Answer) The sentence, “ The listed autophagy inducers-mediated depigmentation is inhibited by knockdown of autophagy essential genes, such as ATG5 [87-90], ATG7 [91], and LC3 [92], or treatment of autophagy inhibitors, such as 3-MA [88,92-95], hydroxychloroquine [96], bafilomycin A1 [89], and chloroquine [97]. For example,” was added, lin2 204~208, page 7. Each agent or stimulator inducing autophagy-mediated depigmentation and its reported findings were listed in table 1.
Comment 4. improve the discussion among the different results presented.
Answer) Thank you for your important point. I think your point is very important to understand the relationship or cross-talk between melanogenesis and autophagy. But there is little clear evidence for that. We tried to present our opinion about that, line 224~236, page 8. “Melanogenesis is a process to synthesis of melanin in melanosome including melanosome formation and autophagy is a process degrading cellular component. Strangely, proteins essential for autophagy degrading cellular components including melanosome is involved in melanogenesis for de novo synthesis of melanin and melanosome. Although the mechanism to clarify the relationship between the two processes has not been elucidated so far, there is no direct evidence that the autophagy process is essential for melanogenesis. Another possibility is that proteins involved in autophagy may play a role in the signaling for melanogenesis independent of the autophagy process. For example, although knockdown or knockout of ATG7, an essential gene for autophagy, inhibits melanin sythesis through reduction of MITF expression [69,70], knockdown of ULK1, an essential kinase forming autophagy initiation complex, induces melanin synthesis by increasing the expression of MITF [75]. ”
Reviewer 3 Report
Autophagy is important in melanosome homeostasis. However, this topic has not been reviewed well in the literature. This review updates the knowledge on the function of autophagy in melanogenesis regulation. It is well written with informative 3 Figures and 1 Table. I have only minor suggestions for the style of this manuscript.
1) Figures 1-3 are not mentioned in the main text. Please insert "Figure 1" etc in the main text.
2) Style of References is not that for the journal Cells.
3) The position of period should be after reference number. There are many sentences that are not in this style. For example, "vitiligo. [14, 15]" should be "vitiligo [14, 15]." These points should be corrected.
4) Page 7. The title of Section 5 needs to be moved to the left and in bold.
5) Page 7. The title of Table 1 needs to be moved to the left.
6) Page 5. "stage 2" and "stage 3" should be "stage II" and stage III".
Author Response
Comment 1. Figures 1-3 are not mentioned in the main text. Please insert "Figure 1" etc in the main text.
Answer) Thank you for your kindness. We cited figures in the text. Figure 1 in line 62, page 2. Figure 2 in line 123, page 4. Fig3 in line 141, page 5. Fig3.1 in line 165, page 5. Fig 3.2 in line 173, page 5. Fig 3.3 in line 183, page 5.
Comment 2. Style of References is not that for the journal Cells.
Answer) Thank you for your kindness. Perhaps there was a mistake during the editing. We corrected the mistake.
Comment 3. The position of period should be after reference number. There are many sentences that are not in this style. For example, "vitiligo. [14, 15]" should be "vitiligo [14, 15]." These points should be corrected.
Answer) As your advice, the position of the period is after Ref number.
Comment 4. The title of Section 5 needs to be moved to the left and in bold.
Answer) Thank you for your kindness. Perhaps there was a mistake during the editing. We corrected that mistake.
Comment 5. Page 7. The title of Table 1 needs to be moved to the left.
Answer) Thank you for your kindness. Perhaps there was a mistake during the editing. We corrected that mistake.
Comment 6. Page 5. "stage 2" and "stage 3" should be "stage II" and stage III".
Answer) Thank you for your kindness. “stage 2” and “stage 3” were changed with “stage II” and “stage III”, respectively, on page 5.
Round 2
Reviewer 2 Report
no comments